# Sponge Whirl-Pak Sampling Method and Droplet Digital RT-PCR Assay for Monitoring of SARS-CoV-2 on Surfaces in Public and Working Environments

**DOI:** 10.3390/ijerph19105861

**Published:** 2022-05-11

**Authors:** Davide Cardinale, Maria Tafuro, Andrea Mancusi, Santa Girardi, Federico Capuano, Yolande Thérèse Rose Proroga, Federica Corrado, Jacopo Luigi D’Auria, Annachiara Coppola, Giuseppe Rofrano, Palmiero Volzone, Pio Galdi, Sabato De Vita, Alfonso Gallo, Elisabetta Suffredini, Biancamaria Pierri, Pellegrino Cerino, Maria Morgante

**Affiliations:** 1Centro di Referenza Nazionale per l’Analisi e Studio di Correlazione tra Ambiente, Animale e Uomo, Istituto Zooprofilattico Sperimentale del Mezzogiorno, Via Salute n. 2, 80055 Portici, Italy; davide.cardinale@izsmportici.it (D.C.); maria.tafuro@izsmportici.it (M.T.); federica.corrado@izsmportici.it (F.C.); jacopoluigi.dauria@izsmportici.it (J.L.D.); annachiara.coppola@izsmportici.it (A.C.); giuseppe.rofrano@izsmportici.it (G.R.); palmiero.volzone@izsmportici.it (P.V.); pio.galdi@izsmportici.it (P.G.); sabato.devita@izsmportici.it (S.D.V.); alfonso.gallo@izsmportici.it (A.G.); strategia@izsmportici.it (P.C.); 2Department of Food Security Coordination, Istituto Zooprofilattico Sperimentale del Mezzogiorno, Via Salute n. 2, 80055 Portici, Italy; andrea.mancusi@izsmportici.it (A.M.); santa.girardi@izsmportici.it (S.G.); federico.capuano@izsmportici.it (F.C.); yolande.proroga@izsmportici.it (Y.T.R.P.); 3Department of Food Safety, Nutrition and Veterinary Public Health, Istituto Superiore di Sanità, Viale Regina Elena, 299, 00161 Rome, Italy; elisabetta.suffredini@iss.it; 4Azienda Sanitaria Locale Avellino, 83100 Avellino, Italy; direzionegenerale@aslavellino.it

**Keywords:** SARS-CoV-2, COVID-19, sponge, surfaces, ddPCR, fomites

## Abstract

The SARS-CoV-2 can spread directly via saliva, respiratory aerosols and droplets, and indirectly by contact through contaminated objects and/or surfaces and by air. In the context of COVID-19 fomites can be an important vehicle of virus transmission and contribute to infection risk in public environments. The aim of the study was to analyze through surface sampling (sponge method) the presence of SARS-CoV-2 in public and working environments, in order to evaluate the risk for virus transmission. Seventy-seven environmental samples were taken using sterile sponges in 17 animal farms, 4 public transport buses, 1 supermarket and 1 hotel receptive structure. Furthermore, 246 and 93 swab samples were taken in the farms from animals and from workers, respectively. SARS-CoV-2 detection was conducted by real-time RT-PCR and by digital droplet RT-PCR (dd RT-PCR) using RdRp, gene E and gene N as targets. None of the human and animal swab samples were positive for SARS-CoV-2, while detection was achieved in 20 of the 77 sponge samples (26%) using dd RT-PCR. Traces of the RdRp gene, gene E and gene N were found in 17/77 samples (22%, average concentration 31.2 g.c./cm^2^, range 5.6 to 132 g.c./cm^2^), 8/77 samples (10%, average concentration 15.1 g.c./cm^2^, range 6 to 36 g.c./cm^2^), and in 1/77 (1%, concentration 7.2 g.c./cm^2^). Higher detection rates were associated with sampling in animal farms and on public transport buses (32% and 30%) compared to the supermarket (21%) and the hotel (no detection). The result of the study suggests that the risk of contamination of surfaces with SARS-CoV-2 increases in environments in which sanitation strategies are not suitable and/or in highly frequented locations, such as public transportation. Considering the analytical methods, the dd RT-PCR was the only approach achieving detection of SARS-CoV-2 traces in environmental samples. Thus, dd RT-PCR emerges as a reliable tool for sensitive SARS-CoV-2 detection.

## 1. Introduction

On 31st December 2019, Chinese health authorities reported a number of unusual cases of pneumonia of unknown origin to the World Health Organization (WHO) in the town of Wuhan, Hubei Province, China [1]. Most of the cases were epidemiologically associated with a market of live animals (Huanan South China Seafood Market), suggesting a possible zoonotic origin (transmission from animals to humans [2]). On 7 January 2020, a new beta-coronavirus was detected through the use of unbiased sequencing in samples from patients with pneumonia. Subsequently, the Chinese authorities reported the identification of a new virus belonging to the same coronavirus family as “Severe Acute Respiratory Syndrome Coronavirus 2” (SARS-CoV-2) [3]. Because of the rapid spread of the virus, on 11 February 2020, the World Health Organization (WHO) and the local Chinese authorities announced that a human-to-human transmission had occurred. On 11 March 2020, the WHO declared the COVID-19 outbreak a global pandemic [4]. Epidemiological data suggest that SARS-CoV-2 can spread directly via saliva, respiratory aerosol and droplets, and indirectly by contact through contaminated objects and/or surfaces. The person-to-person spreading of SARS-CoV-2 occurs mainly through respiratory droplets, which a patient expels when coughing, sneezing, or speaking. The SARS-CoV-2 virus is able to remain intact in droplets suspended in the air for up to three hours [5,6,7]. Beside direct transmission through the contact between mucous membranes (mouth, eyes, nose) and infected droplets, SARS-CoV-2 can infect humans indirectly, when hands come to contact with mucosae after touching an infected person or contaminated objects [8]. As underlined by Kraay et al. (2018), the fomites have a crucial role as potential transmission vehicles of respiratory viruses or other pathogens [9,10]. In particular, the transmission mediated by fomites depends on behavioural factors (i.e., hand hygiene practices), on the physical environment (i.e., factors affecting virus viability) and on the type of surfaces that may undergo contamination. The dynamics in fomites contamination and transmission appear to be very complex as it involves multiple variables.

Studies conducted over the last two years, indeed, highlighted the impact of factors such as temperature, surface porosity, relative humidity, air drying, and direct light on SARS-CoV-2 infectivity retention, either on surfaces or in air [11,12,13,14,15,16]. In addition to this, the different persistence of the virus on surfaces of different natures (plastic, cardboard, metal, etc.) have been assessed [17,18,19,20] and predictive tools and mathematical models have been developed to calculate virus decay on surfaces based on multiple factors [21]. In the course of the COVID-19 pandemic, fomites have been considered a relevant vehicle of virus transmission, contributing to the risk of infection in public environments such as schools, long-term care facilities and hospitals [22,23,24,25]. In this context, several studies have addressed the detection of SARS-CoV-2 on inanimate surfaces in close contact with COVID-19 patients, either in hospital and healthcare settings [26,27,28,29], in households and isolation rooms [30,31,32] or in specific settings such as a cruise ship during outbreak events [33]. Less investigated, on the contrary, has been the presence of SARS-CoV-2 (either genome or infectious virus) in public spaces or in other working environments. A study performed by Silva and colleagues (2021) [34] on public urban areas in Recife, Brazil showed high positivity rates in public transportation terminals (>50%), public parks, supply centers, and markets, with toilets, bancomat, handrails, playgrounds, and other highly touched public surfaces being frequently positive for SARS-CoV-2. Similarly, sampling on public urban surfaces (crosswalk buttons, trash can handles, door handles of essential commercial services such as grocery shops, banks, gas stations, etc.) in Somerville, Massachusetts, USA between March and June 2020, achieved the detection of SARS-CoV-2 in 8.3% of samples [35] and sampling in playgrounds and on water fountains in six cities in Israel in August 2020 showed the presence of viral genome in ~4% of the samples [36]. Detection of SARS-CoV-2 genetic material on public transportation has also been addressed by several authors, with results ranging from a 17% positivity rate in transports in Quito, Ecuador [37] to 43% in Barcelona, Spain [38]. In Italy, a first study on public transportation was performed in the city of Chieti (Abruzzo region, central Italy) in May 2020 with no detection of SARS-CoV-2 RNA on surfaces [39], but a more extensive assessment of SARS-CoV-2 deposition on surfaces in public places/services has been performed in the region of Apulia, where virus presence has been investigated in supermarkets (detection in 4.3% of samples), public transportation, such as trains and buses (10.7% of positive results), and tourist recreational facilities (virus detection in 6% of swabs) [40,41,42], but no data are available for other kind of environments and for other areas of the country. These studies consider public transportation in urban areas (with high population density conditions). Moreover, the observed differences in the percentage of positivity rate could also depend on (i) the time when the study was conducted (i.e., during the first wave of contagion, March–April 2020, Italian public transportation were not frequented and, with major turnout in northern Italy rather than southern); (ii) the levels of viruses spreading at the time of surfaces sampling; (iii) the different design of the study, including the samples collection methods. Considering the analytical aspect, the current gold standard for the detection and diagnosis of SARS-CoV-2 infection is based on the real-time RT-PCR. The low levels of virus contamination are the main obstacles for the application of RT-PCR for the detection of viruses on surfaces. Furthermore, sensitivity of the method is not always enough to detect the low viral concentration. Droplet digital PCR (ddPCR) is an alternative PCR method that allows absolute quantification and is more precise than standard PCR. These properties make ddPCR a promising SARS-CoV-2 detection method for samples with low template.

The purpose of the present study was to investigate the presence of SARS-CoV-2 RNA on surfaces of widely different environments: (I) two types of public environments, buses for public transportation and a supermarket; (II) a receptive structure involved in a SARS-CoV-2 outbreak associated to a wedding party; (III) 17 farms in the territory of the Campania region chosen based on the detection of a SARS-CoV-2 infection among the employees. 

We focused our attention on disparate environments, and the criteria of selection was based on the perceived risks (no medical areas, but highly frequented work environments without standard protocol of biohazard management for SARS-CoV-2 pandemic).

## 2. Materials and Methods

### 2.1. Sampling

Sampling was carried out between 14 August 2020 and 14 April 2021 in 17 farms, 4 public transportation buses, 1 supermarket and 1 hotel receptive structure distributed in the territory of the Campania region (Figure 1).

Farm sites were chosen due to connection to a previously notified case of SARS-CoV-2, involving either employees or visitors. For 6 of the 17 farms investigated, in the two months preceding sampling, five COVID-19 cases (and two fatalities) were reported in workers (one of them being a regular visitor of two farms). Based on available records, farms were managed in agreement with legislation on hygiene, animal welfare and biocontainment (where relevant), but no specific protocol besides general hygiene, distancing and face mask rules was implemented for SARS-CoV-2 in these working environments. The 17 animal farms included: buffaloes (*n* = 4), cattle (*n* = 1), goats and cattle (*n* = 1), goats (*n* = 2), hens (*n* = 2), horses (*n* = 2), pigs (*n* = 3), pigeons (*n* = 1), and rabbits (*n* = 1), and sampling points included milking parlor surfaces, barkers, stable walls.

The 4 public transportation buses and the supermarket were subjected to daily sanitization practices, in agreement with recommendations implemented during the COVID-19 epidemic, and sampling was performed after cleaning and before the service. Sampling points on public transportation buses included handrails, stop request buttons, walls, handles, holding supports, stamping machines, etc., and trolley and cart handles, fridge/freezer handles, weighing scales, card readers, etc. in the supermarket.

Finally, the hotel receptive structure—also operating under the guidelines for SARS-CoV-2 transmission containment—was included in the study following its involvement in a SARS-CoV-2 outbreak (20 cases infected during a wedding party) and the environmental sampling of the surfaces bar countertop, tap, kitchen table, fridge and table, dining room tables was carried out one week after the event and after sanitization.

Sponge swipe was conducted using a 3.8 × 7.6 × 1.5 cm sterile sponge (VWR International, Radnor, PA, USA) moistened with sterile peptone water (LabRobot products, Stenungsund, Sweden). During the sampling procedure, the area to be sampled (100 cm^2^) was identified, the bag containing the dehydrated sponge was opened and 10 mL of diluent was added to hydrate the sponge. Subsequently, the sponge was vigorously swiped on the previously defined area, first with horizontal and then with vertical movements. At the end of sampling, the sponge was placed back in its own bag, sealed by rolling the upper edge several times and placed in an insulated container equipped with dry ice packs. Transport to the laboratory for analysis was completed as soon as possible and in any case within 24 h of sampling.

To better understand the role of the environment in potential virus transmission in a complex environment such as an animal farm, each environmental (sponge) sampling was associated with sampling on human subjects (workers of the farms) and on animals. For the humans, nasopharyngeal swabs were performed, under the authorization of the GENCOVID study protocol “Health surveillance for the SARS-CoV-2 virus, responsible of COVID-19 pandemic in high-risk population or in people in direct contact with positive patients” (approved by the ethics committee of University of Naples “Federico II”, approval n° 141/20). For animals, either nasopharyngeal (buffaloes, pigs, rabbits, horses, goats, cattle) or oropharyngeal (hens and pigeons) swabs were performed. Both human and animal samplings were performed using the Copan UTM-RT system (Universal Transport Medium for Viruses) with FLOQSwabs (Copan Diagnostics, CA, USA). Samples were transported to the laboratory with triple packaging under controlled temperature (4 °C).

Overall, a total of 77 environmental samples (sponge—S), 246 animal swabs (AS) and 93 human swabs (HS) (Table 1) were taken, divided as follows: 12 AS, 2 HSand 10 S in the province of Naples, 97AS, 39HSand 13 S in the province of Caserta, 100AS, 44HSand 51 S in the province of Salerno, 16AS, 6HSand 1 S in the province of Benevento, 21AS, 2 HSand 2 S in the province of Avellino.

### 2.2. Nucleic Acids Extraction

For nucleic acids extraction from environmental samples, 20 mL Proteinase K solution (100 µg mL^−1^) (Qiagen, Hilden, Germany) were added to sponge bags. Extraction efficiency was monitored by the addition of 30 µL of internal process control (IPC) virus (Mengo virus strain MC_0_). Samples were incubated at 37 °C for 60 min, followed by 15 min at 60 °C and all liquid was recovered squeezing the sampling sponge. Finally, after centrifugation at 3000× *g* for 5 min to sediment debris, the liquid was collected, measured (on average ≈20 mL), and retained for RNA extraction. Total RNA was extracted from 500 µL of the liquid using the eGene-UP semi-automated system (bioMerieux, Marcy-l’Étoile, France) and NucliSENS buffer reagents (bioMerieux) following the manufacturer’s instructions. Elution was performed in 100 µL. A negative extraction control sample (molecular grade water) was also tested in parallel with each set of extracted samples.

Nucleic acid extraction from human and animal swabs was performed using an automated nucleic acid platform (Maelstrom 9600, TANBead, Taiwan), with a magnetic bead-based protocol, using an extraction associated kit (TANBeadNucleic Acid Extraction Kit). Following manufacturer’s instructions, 300 µL of virus transport medium for the swabs were used for nucleic acids extraction. The RNA was finally eluted in 80 µL of elution buffer in the dedicated plate provided with the kit.

Nucleic acids were stored at −80 °C until testing.

### 2.3. SARS-CoV-2 Detection by Real-Time RT-PCR

Viral extraction efficiency in sponge samples was assessed using Mengovirus according to ISO 15216-2:2019 [43]. SARS-CoV-2 detection was performed following the Charité-Berlin protocol, both for primers/probes and for thermal profile [44] for the amplification of the SARS-CoV-2 E (Betacoronavirus screening assay), N (nucleocapsid protein) and RdRp (SARS-CoV-2 confirmatory assay) genes (Table 2).

Amplification reactions were prepared with the RNA UltraSense One-Step qRT-PCR System (Invitrogen; Waltham, MA, USA) and presence of PCR inhibitors was ruled out using TaqMan Exogenous Internal Positive Control (applied Biosystems; Waltham, MA, USA) according to manufacturer’s instructions. Results were considered acceptable if a minimum recovery of 1% was achieved and no significant inhibition was detected in the amplifications. SARS-CoV-2 detection in human and animal swabs was performed using the Allplex 2019-nCoV real-time RT-PCR assay (Seegene, South Korea) which also targets the E gene of Sarbecovirus, and the RdRp and N genes of SARS-CoV-2. An internal control (MS2 phage genome), provided with the kit, was used to monitor the process. Amplification and detection were performed on Bio-Rad CFX96 platform (Bio-Rad, Hercules, CA, USA) and results were interpreted according to the manufacturer instructions as follows: a sample was considered (I) positive if the three target genes displayed *Ct* values < 38, (II) presumptive positive if one or two genes were amplified with *Ct* values >38, (III) negative, if no amplification was detected for any of the three targets or if the three genes displayed *Ct* values ≥38.

### 2.4. SARS-CoV-2 Detection in Environmental Samples by Droplet Digital RT-PCR (dd RT-PCR)

Droplet digital RT-PCR was performed on Bio-Rad’s QX200 system (Bio-rad, Hercules, CA, USA) using the primers/probes of the Charité-Berlin protocol [18]. Two probes were used to amplify the RdRp gene at the same time: RdRP_SARSr-P1to detect SARS-CoV-2 and bat-SARS-related CoVs and RdRp_SARSr-P2 for specific detection of SARS-CoV-2. The reaction mixture (20 μL of total volume) consisted of: 5 μL One-step RT-ddPCR Advanced Kit for Probes, 2 μL Reverse transcriptase, 1 μL 300 mM DTT, 5 μL of samples RNA, primers and probes in the concentrations detailed in Table 2 and nuclease-free water as requied. Droplet generation was then performed as recommended by the manufacturer, and amplification was carried out on a CFX96 DeepWell instrument (Bio-Rad) with the following thermal profile: 50 °C for 60 min, 95 °C for 10 min followed by 95 °C for 15 s and 60 °C for 45 s (45 cycles) and by a final stage at 98 °C for 10 min. After amplification, results were acquired using the Bio-Rad QX200 Droplet Reader and QuantaSoft software (v1.7) (Bio-rad, Hercules, CA, USA) was used to count the PCR-positive or PCR-negative droplets and to provide absolute quantification of the target sequence. For each target gene, results of dd RT-PCR were used to calculate positivity rate (number of samples with SARS-CoV-2 detection on tested samples) for the different environments considered in the study, and average SARS-CoV-2 concentrations and range in positive samples. Detection rates in the different types of environments (animal farms, public transportation, supermarkets) were compared with the chi-squared test.

## 3. Results

In the present study, we investigated the presence of SARS-CoV-2 virus in public (transportation, supermarket, hotel) and working environments (animal farms) associated with reported COVID-19 cases in different areas of the Campania region. In animal farms, human and animal naso/oropharyngeal swabs were also taken. None of the naso/oropharyngeal swabs from humans (*n* = 93) and animals (*n* = 246) showed the presence of any of the three genetic markers (RdRp, E and N gene) of SARS-CoV-2. With regard to environmental samples (sponges), SARS-CoV-2 detection by real-time RT-PCR was also negative for all targets, in all samples (data not shown). On the contrary, dd RT-PCR achieved the detection of traces of SARS-CoV-2 RNA in several samples (Table 3).

Considering the three genetic targets (RdRp, gene E, gene N) together, 20/77 samples (26%) displayed the presence of at least one marker. In detail, RdRp gene sequences were detected in 17/77 samples (22%, concentration range 0.14–3.30 genome copies (g.c.)/µL of RNA), gene E in 8/77 (10%, concentration range 0.15–0.90 g.c./µL) and gene N in 1/77 (1%, concentration 0.18 g.c./µL). These results corresponded to estimated concentrations of SARS-CoV-2 of 31.2 g.c./cm^2^, range 5.6 to 132 g.c./cm^2^ considering RdRp, 15.1 g.c./cm^2^, range 6 to 36 g.c./cm^2^ using gene E, and 7.2 g.c./cm^2^ considering gene N (data not shown).

SARS-CoV-2 detection was more frequent in samples from animal farms (11/34 = 32% positive samples; positivity rate of 29%, 15% and 0% for RdRp, gene E and gene N, respectively) and in public transportation buses (6/20 = 30% positive samples; 30% RdRp, 10% gene E and 0% gene N), while each of the three SARS-CoV-2 targets were detected only in 1 of the 14 samples from supermarket (positivity rate of 7%). In these samples, however, a higher divergence of results was found, as the three targets were never detected in association but in three distinct samples. The differences in the detection rates of SARS-CoV-2 in animal farms, public transportation, and supermarkets were not statistically significant (chi-squared *p*-value = 0.75).

Also, virus concentration was higher in samples collected from animal farms compared to other sampling locations, RdRp average value being 1.00 g.c./µL of RNA (equivalent to 40 g.c./cm^2^, range 5.6 to 132) vs. 0.51 g.c./µL of RNA (equivalent to 20.4 g.c./cm^2^, range 7.2 to 40) of samples from public buses. Finally, none of the nine samples collected from the receptive structure of the hotel was found to harbor genetic traces of SARS-CoV-2.

## 4. Discussion

The SARS-CoV-2can survive in the air and on surfaces for several hours and up to several days, depending on environmental conditions [45], but being infected following contact with contaminated surfaces seems to be a remote risk [46]. Despite this, public health agencies underline that surfaces may pose a risk for virus transmission and should be disinfected frequently, prioritizing efforts to prevent the spread of the virus [47]. The objective of our study was to characterize SARS-CoV-2 contamination in selected public and working environments, using the sponge method for surface sampling and droplet digital PCR for detection [48,49]. To date, indeed, several studies have assessed SARS-CoV-2 contamination in hospital or healthcare settings, but still few have addressed the risk of surface contamination in public- or work- places [38]. Environmental sampling is a critical component of verification of the effectiveness of sanitation processes and, in order to detect the presence of viruses that may be present in traces on the investigated surfaces, the use of efficient sampling systems is a compelling requirement [50]. The sponge sampling method allows a more vigorous scrubbing of the surfaces, an extensive adhesion of the sampler to the surface (also on objects with uneven shapes or with crevices), and an increased adsorption of detached materials. Over time, swabs with higher recovery performance towards viruses on surfaces [51] have been developed, through adaptation of shape and materials, and have also been successfully used for SARS-CoV-2 sampling from plastic objects such as glasses, cups and bottles that may come into contact with infected individuals [52]. However, in the context of official controls performed by the local health authorities, it has long been demonstrated that swabs have a capture capacity not exceeding ~20% of the target on the surface, therefore the sponge sampling confirms its functionality because of the higher recovery efficiency [53,54,55,56,57,58]. Further to an effective sampling strategy, high sensitivity should also be provided by the molecular analytical methods used for testing of environmental samples. The current gold standard for the detection and diagnosis of SARS-CoV-2 infection is based on the real-time RT-PCR assay but has several downsides [59]. False-negative real-time RT-PCR assay results can arise because of insufficient viral material in samples, while weak positive results (i.e., high *Ct* values) can be difficult to interpret and to differentiate from technical artifacts (e.g., fluorescence increase due to sample background or probe degradation) [44]. For this reason, in recent years, droplet digital RT-PCR (dd RT-PCR) has been considered a good alternative to real time RT-PCR assays, considering its high sensitivity in the detection and quantification of RNA targets [60,61] and its ability to provide consistent and reliable results also at low target concentrations [62]. Moreover, using Poisson’s distribution, the absolute concentration of the target can be calculated without the use of standard curves [63] and, due to sample partitioning, reactions are less affected by the presence of inhibitors [64]. Therefore, in this study, sponge sampling was coupled with real time RT-PCR as well as with dd RT-PCR, which, in the end, was the only analytical approach providing positive results. Our investigation showed traces of SARS-CoV-2 in 26% of the surface samples collected from the different environments considered in the study (animal farms, public transportation buses, supermarkets, hotel receptive structures), some of which interested from previous outbreak events. Overall, considering the results of the RdRp gene as a reference, SARS-CoV-2concentration in the positive samples was of 31.2 g.c./cm^2^(range 5.6 to 132 g.c./cm^2^). Quantitative data on SARS-CoV-2 contaminated surfaces are scarce and are almost exclusively related to high risk closed environments used for COVID-19 patients [29], therefore direct comparison of our results, which derive from different environments, with other studies is not feasible. With regard to sampling performed on public transportation buses, positive results (always considering the RdRp gene as a reference) were obtained in 30% of samples, with an average concentration of 40 g.c./cm^2^(range 5.6 to 132g.c./cm^2^). These results are intermediate to those stated by studies from other countries [37,38] and only slightly higher than those reported for the period May–June 2021 in another Italian region, Apulia, in which SARS-CoV-2 was overall detected on 10.7% of the tested surfaces and, more specifically, in 19.3% of bus surfaces and 2% of train surfaces [42]. It should be noted, however, that the different prevalence might also be related to the different sampling techniques (swabs vs. sponges), which—as stated before—might affect viral recovery from surfaces. Interestingly, the viral concentrations achieved in the present study seem significantly higher than those reported in one [38] of the few other studies including quantitative data. Indeed, in the study by Moreno and colleagues (2021) [38] on public transportation in Barcelona, depending on the target sequence (IP2, IP4, gene E), SARS-CoV-2 contamination ranged from 5 to 490 c.g./m^2^ in public buses and from 18 to 714 c.g./m^2^ in the subway, i.e., approximately 4 log lower than in our study. This significant divergence in quantitative results, however, should be considered in the light of both the deep difference in the analytical methods (sampling techniques, molecular targets, real-time PCR vs. dd RT-PCR, standards for calibrations, etc.) and the different epidemiological situations registered during the two studies (May–June 2020 for the study in Barcelona, March 2021 for the current study). Considering the samples taken from the supermarket, a certain variability of results was observed, as overall 3 samples of 14 (21%) showed traced of SARS-CoV-2 genome. These results are higher than those reported by Caggiano and colleagues [40] for supermarkets tested in Apulia, Italy in April–May 2021 (4.3%) or by Singh et al. [65] in Ontario, Canada during October–November 2020 (no detection of SARS-CoV-2). However, the three studies differ under several aspects, including different sampling techniques (swab vs. sponges), molecular detection (real-time vs. dd RT-PCR), as well for the fact that only this study targets multiple genome regions of SARS-CoV-2, hence increasing the chance of detection. It should also be noted that the three positive results of the present study were all associated only with one genetic target (none of the three tested genes were ever detected simultaneously). This could be due to high viral degradation, possibly in relation to the stricter cleaning and sanitation procedures applied in food stores. Indeed, while detection of SARS-CoV-2 is a clear indication of a contamination event, little can be said on viral infectivity particularly if detection is related to surfaces periodically subjected to disinfection.

Effectiveness of cleaning and sanitation procedures (for the deactivation of SARS-CoV-2 as well as for removal of any residual inactivated virus particle), together with the natural decay of viruses on surfaces, may also explain the results obtained in the samples collected from the hotel receptive structure, which was sampled due to its association with a SARS-CoV-2 outbreak event following a wedding party. Indeed, in this case sampling was performed one week after the event and after extensive sanitation. Sampling of receptive structures in a different context (e.g., under routine use of the environments and with standard cleaning activities) may indeed lead to SARS-CoV-2 detection also in such environments, as demonstrated by Montagna and colleagues (2021) [41], that found traces of the viral genome in 6% of the tested surfaces. Finally, with regard to the samples taken in animal farms (buffaloes, cattle, pigs, goats, horses, rabbits, hens and pigeons), 32% of surface samples were found positive for at least one genetic target of SARS-CoV-2. With the exception of studies dealing with mink farms, in which SARS-CoV-2 on surfaces was investigated due to the outbreaks associated with the CoV-2 mink-specific variant [66,67], the present study is, to our knowledge, the first one dealing with environmental contamination by SARS-CoV-2 in animal farms. Significantly, the absence of SARS-CoV-2 detection in the workers and in the animals at the moment of testing (no detection in 339 swabs) confirms that the 11 positive surface samples were related to previous infection events, during which the virus had been deposited on work surfaces, walls and utensils. Further to this, higher viral concentrations were found in samples taken from animal farms (40 g.c./cm^2^ vs. 20.4 g.c./cm^2^ of public transportation, with a range significantly more extended, 5.6–132 vs. 7.2–40). This is in agreement with the fact that cleaning and sanitation procedures are less feasible in animal husbandry, and therefore removal of viruses deposited on surfaces is more difficult. One of these, buffaloes farm n°4, provided three positive samples (two of which from the milking parlor surfaces) and had registered two COVID-19 fatalities among workers in the two months preceding our sampling. The other two farms, buffaloes farm n°2 and n°3, also regularly visited by one of these workers, showed instead no traces of SARS-CoV-2 on the tested surfaces, hence displaying a certain variability of viral deposition on surfaces, depending on the environment.

## 5. Conclusions

Data obtained suggests that the risk of contamination increases in environments with a higher presence of people and where decontamination strategies are not always feasible. At the same time our observations also suggest that improving the quality of the environmental sanitation can reduce the spreading of the pathogen. Comparing data on human swabs and on surface sponges, it has been previously shown that, after disinfection, several environments, even those affected by proven human contamination, do not present traces of SARS-CoV-2, which instead are revealed in different surfaces collected from environments not thoroughly sanitized [68]. A rapid SARS-CoV-2 inactivation is possible by using commonly available chemicals and biocides on inanimate surfaces (e.g., EtOH 70% or sodium hypochlorite) [69]. Further to this, this study showed the reliability of dd RT-PCR method for the detection of SARS-CoV-2 traces on surfaces. Further studies are required to improve and standardize surface-sampling methods for viral analysis, to estimate viral infectivity on surfaces, and to assess the risk of viral transmission, thereby providing additional information on the relevance of SARS-CoV-2 contamination of fomites.

## Figures and Tables

**Figure 1 ijerph-19-05861-f001:**
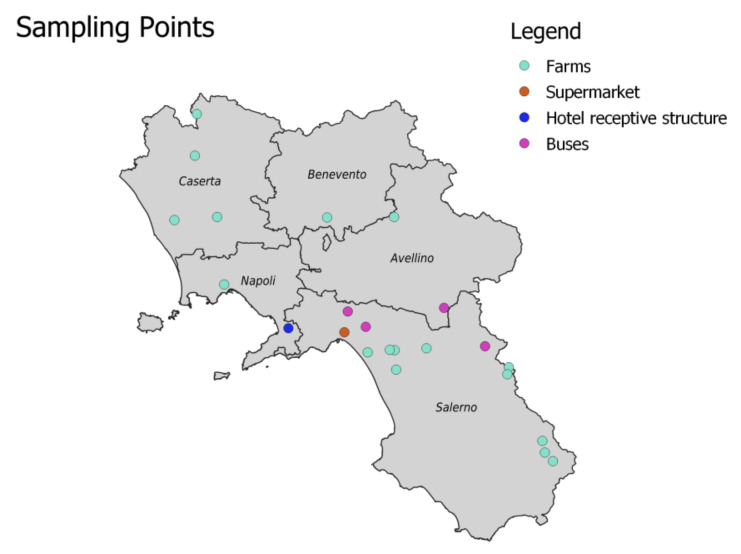
Sample distribution on the territory of Campania.

**Table 1 ijerph-19-05861-t001:** Samples analyzed in the study, divided by sampling location.

Sampling Location	District	Environmental Samples	Animal Swabs	Human Swabs
Buffaloes farm_1	Caserta	1	20	7
Buffaloes farm_2	Caserta	1	22	11
Buffaloes farm_3	Caserta	1	23	12
Buffaloes farm_4	Caserta	10	32	9
Cattle farm	Salerno	-	10	1
Goat and cattle farm	Salerno	2	17	2
Goat farm_1	Salerno	1	6	-
Goat farm_2	Salerno	-	10	1
Hen farm_1	Salerno	1	10	5
Hen farm_2	Salerno	1	11	10
Horse stall_1	Salerno	4	6	3
Horse stall_2	Salerno	6	14	4
Pig farm	Salerno	1	9	7
Pig farm	Benevento	1	16	6
Pig farm	Avellino	2	21	2
Pigeon farm	Napoli	1	12	2
Rabbits farm	Salerno	1	7	11
Bus_1	Salerno	5	-	-
Bus_2	Salerno	5	-	-
Bus_3	Salerno	5	-	-
Bus_4	Salerno	5	-	-
Supermarket	Salerno	14	-	-
Hotel receptive structure	Napoli	9	-	-
**Total**		**77**	246	93

**Table 2 ijerph-19-05861-t002:** Sequence and concentrations of primers and probes used for real-time RT-PCR and Droplet Digital-RT-PCR detection of SARS-CoV-2 in environmental samples.

Primers and Probes	Sequence	Concentrations
RdRp_SARSr-F	GTGARATGGTCATGTGTGGCGG	600 nM
RdRp_SARSr-R	CARATGTTAAASACACTATTAGCATA	800 nM
RdRP_SARSr-P1	FAM-CCAGGTGGWACRTCATCMGGTGATGC-BBQ	100 nM
RdRp_SARSr-P2	FAM-CAGGTGGAACCTCATCAGGAGATGC-BBQ	100 nM
E_Sarbeco_F	ACAGGTACGTTAATAGTTAATAGCGT	400 nM
E_Sarbeco_R	ATATTGCAGCAGTACGCACACA	400 nM
E_Sarbeco_P1	FAM-ACACTAGCCATCCTTACTGCGCTTCG-BBQ	200 nM
N_Sarbeco_F	CACATTGGCACCCGCAATC	600 nM
N_Sarbeco_R	GAGGAACGAGAAGAGGCTTG	800 nM
N_Sarbeco_P	FAM-ACTTCCTCAAGGAACAACATTGCCA-BBQ	200 nM

FAM: 6-carboxy-fluorescein; BBQ: BlackBerry Quencher.

**Table 3 ijerph-19-05861-t003:** Detection of SARS-CoV-2 marker genes in environmental samples by droplet digital RT-PCR.

Sampling Location	Municipality	N° of Samples	dd RT-PCR		
			RdRp	E Gene	N Gene
Buffaloes farm_1	Caserta	1	**+**	-	-
Buffaloes farm_2	Caserta	1	-	-	-
Buffaloes farm_3	Caserta	1	-	-	-
Buffaloes farm_4	Caserta	1	**+**	-	-
Buffaloes farm_4	Caserta	2	**+**	**+**	-
Buffaloes farm_4	Caserta	7	-	-	-
Cattle farm	Salerno	1	-	-	-
Goat and cattle farm	Salerno	1	-	-	-
Goat farm_1	Salerno	1	-	-	-
Goat farm_2	Salerno	1	**+**	-	-
Hen farm_1	Salerno	1	**+**	-	-
Hen farm_2	Salerno	1	-	-	-
Horse stall_1	Salerno	2	-	-	-
Horse stall_1	Salerno	2	**+**	**+**	-
Horse stall_1	Salerno	2	**+**	-	-
Horse stall_2	Salerno	4	-	-	-
Pig farm	Salerno	1	-	-	-
Pig farm	Benevento	1	-	-	-
Pig farm	Avellino	1	-	-	-
Pigeon farm	Napoli	1	-	-	-
Rabbits farm	Salerno	1	-	**+**	-
** *subtotal* **		**34**	**10 (29%)**	**5 (15%)**	**0 (0%)**
Bus_1	Salerno	3	-	-	-
Bus_1	Salerno	1	**+**	-	-
Bus_1	Salerno	1	**+**	**+**	-
Bus_2	Salerno	1	**+**	-	-
Bus_2	Salerno	4	-	-	-
Bus_3	Salerno	1	**+**	**+**	-
Bus_3	Salerno	3	-	-	-
Bus_3	Salerno	1	**+**	-	-
Bus_4	Salerno	4	-	-	-
Bus_4	Salerno	1	**+**	-	-
** *subtotal* **		**20**	**6 (30%)**	**2 (10%)**	**0 (0%)**
Supermarket	Salerno	11	-	-	-
Supermarket	Salerno	1	-	**+**	-
Supermarket	Salerno	1	-	-	**+**
Supermarket	Salerno	1	**+**	-	-
** *subtotal* **		**14**	**1 (7%)**	**1 (7%)**	**1 (7%)**
Hotel receptive structure	Napoli	9	-	-	-
** *subtotal* **		**9**	**0 (0%)**	**0 (0%)**	**0 (0%)**
**Total**		**77**	**17 (22%)**	**8 (10%)**	**1 (1%)**

## Data Availability

The data presented in this study are available on request from the corresponding author.

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
