# Peer review of "Sponge Whirl-Pak Sampling Method and Droplet Digital RT-PCR Assay for Monitoring of SARS-CoV-2 on Surfaces in Public and Working Environments"

_ijerph, 2022, doi:10.3390/ijerph19105861_

Round 1

Reviewer 1 Report

This paper aims to look at environmental contamination rates of SARS-CoV2 and how this could be important in fomite transmission. 

Introduction

  1. Please mention in the introduction about ddPCR as this is one of your aims to evaluate this technology in identifying environmental contamination something which is missed and likewise please in table 2 explain the acronyms in the table figure. 
  2. please change sentence 'isolation and identification of the pathogen have been obtained' 
  3. please remove 'after a jump of species form animal to man' and alter sentence
  4.  Please elaborate on the persistence of the virus on different surfaces as this is very much linked to your results and contextualises them.
  5. Please remove the link before reference 21
  6. There is obviously a lot of confounders why there is different positivity rates between different countries and environments please discuss this further including were these studies in the urban or rural setting
  7. Please address your aims and consider why did you pick these disparate environments. Why farms ? 
  8. Would be useful to evalaute different diagnostic platforms sensitivities etc in the environment is there anything that can be learnt from inpatient environmental contamination.

Methods

Well written for the most part

  1. Out of the 17 farms why did we think with the farm was the source given the very low numbers. Moreover, you mention no specific protocol beside general hygeine , distancing and face masks were implemented, I am curious as to what other measures you were thinking ? Likewise the animals you tested many are not spill over hosts or susceptible hosts for coronavirus so please explain why they were picked as some have only been shown to be infective after experimental inoculation.   SARS-CoV-2 in animals: From potential hosts to animal models - PMC (nih.gov) 
  2. Can we clarify this minimum recover of 1% as I am not aware of this being a standard

Discussion

1. Relevant to the overall themes of the paper.

Reviewer 2 Report

The study described in the article “Sponge Whirl-Pak Sampling Method and Droplet Digital RT-PCR Assay for Monitoring of SARS-CoV-2 on Surfaces in Public and Working Environments” by Cardinale et al. is very interesting, both from the scientific and practical point of view. Indeed, more investigations on SARSCoV-2 contamination of surfaces in public- or working environments are needed, especially places that are difficult to sanitize. Moreover, it describes an efficient procedure for checking effectiveness of sanitation processes. The use of sponge method for surface sampling and of droplet digital PCR for detection of SARS-CoV-2 make this study original and useful for development of standardized high quality procedures for environmental sanitation.

However, considering the three genetic targets (RdRp, E gene, N gene) used to detect SARS-COV-2 by ddPCR, authors should try to explain/comment why RdRp is detected in more samples than N and E genes. Is the amplification of the three targets equally efficient with ddPCR?

Minor suggestions:

In the Abstract avoid the statement “in order to evaluate the risk for virus transmission”, because this was not the aim of the study and virus transmission from sponge samples was not addressed.

Introduction: please specify the abbreviation ATMs.

Table 1: correct Environmental Aamples

Minor spell check might be needed.

The present manuscript contains all the appropriate chapters, data are well organized and clearly represented.

Reviewer 3 Report

In this study, the authors used sponge to collect samples from surfaces in public places especially the farms to examine the infectiousness of SARS-CoV-2, which has not been described by other groups. They analyzed the samples by real-time RT-PCR and digital droplet RT-PCR (dd RT-PCR) and demonstrated that dd RT-PCR was the only approach achieving detection of SARS-CoV-2 traces in environmental samples. The conclusions were supported by some data. However, the results presented do not convincingly support their conclusions and several issues need to be addressed.

  1. The authors should provide more detailed information about the priority and significance of using sponge to collect samples in the introduction part.
  2. There was no description of statistical analysis in the materials part, which should be added.
  3. Although the authors showed that use of sponge could improve detection, they did not have a control to compare with, such as swap sampling .
  4. The conclusions cannot be justified by the presented data if they cannot exclude the possibility that the sensitivity was due to sponge sampling.
  5. The authors should carefully proof read and correct grammar mistakes and errors in the manuscript.

Round 2

Reviewer 3 Report

In this study, the authors used sponge to collect samples from surfaces in public places especially the farms to examine the infectiousness of SARS-CoV-2, which has not been described by other groups. They analyzed the samples by real-time RT-PCR and digital droplet RT-PCR (dd RT-PCR) and demonstrated that dd RT-PCR was a better approach achieving detection of SARS-CoV-2 traces in environmental samples. The conclusions were supported by solid data. And it may be used widely after further analysis and improvement.